

# Size dimorphism and sexual segregation in pheasants: tests of three competing hypotheses

Mark A. Whiteside[1,2], Jayden O. van Horik[1], Ellis J.G. Langley[1], Christine E. Beardsworth[1] and Joah R. Madden[1]

[1] Centre for Research in Animal Behaviour, University of Exeter, Exeter, Devon, United Kingdom
[2] Game and Wildlife Conservation Trust, Fordingbridge, Hampshire, United Kingdom

## ABSTRACT

Fine scale sexual segregation outside of the mating season is common in sexually dimorphic and polygamous species, particularly in ungulates. A number of hypotheses predict sexual segregation but these are often contradictory with no agreement as to a common cause, perhaps because they are species specific. We explicitly tested three of these hypotheses which are commonly linked by a dependence on sexual dimorphism for animals which exhibit fine-scale sexual segregation; the Predation Risk Hypothesis, the Forage Selection Hypothesis, and the Activity Budget Hypothesis, in a single system the pheasant, *Phasianus colchicus*; a large, sedentary bird that is predominantly terrestrial and therefore analogous to ungulates rather than many avian species which sexually segregate. Over four years we reared 2,400 individually tagged pheasants from one day old and after a period of 8–10 weeks we released them into the wild. We then followed the birds for 7 months, during the period that they sexually segregate, determined their fate and collected behavioural and morphological measures pertinent to the hypotheses. Pheasants are sexually dimorphic during the entire period that they sexually segregate in the wild; males are larger than females in both body size and gut measurements. However, this did not influence predation risk and predation rates (as predicted by the Predation Risk Hypothesis), diet choice (as predicted by the Forage Selection Hypothesis), or the amount of time spent foraging, resting or walking (as predicted by the Activity Budget Hypothesis). We conclude that adult sexual size dimorphism is not responsible for sexual segregation in the pheasant in the wild. Instead, we consider that segregation may be mediated by other, perhaps social, factors. We highlight the importance of studies on a wide range of taxa to help further the knowledge of sexual segregation.

# INTRODUCTION

Sexual segregation, in which females and males are separated in time and/or space outside of the mating season, is common in a variety of birds, mammals, fish and reptiles (*Bleich, Bowyer & Wehausen, 1997*; *Ruckstuhl & Neuhaus, 2005*; *Ruckstuhl & Neuhaus, 2000*). Fine scale sexual segregation, in which segregation occurs within a small area, is particularly

Corresponding author
Mark A. Whiteside,
M.Whiteside@exeter.ac.uk

prevalent in species with strong sexual dimorphism (*Ruckstuhl & Neuhaus, 2002*) and those with polygynous mating systems (*Clutton-Brock, 1989*). However, there is little consensus about the underlying factors driving sexual segregation (*Alves et al., 2013*; *Bonenfant et al., 2007*).

Sexual body-size dimorphism can result in differential: (1) metabolic rates and absolute intake requirements (*Demment & Van Soest, 1985*); (2) ability to digest and absorb nutrients (*Demment & Van Soest, 1985*); (3) physical characteristics (e.g., molar size *Lüps & Roper, 1988*); and (4) parasite infection rates and immuno-competence (*Stoehr & Kokko, 2006*; *Zuk & McKean, 1996*) between the sexes. These differences underlie the four main, but non-exclusive, hypotheses proposed to explain why sexual segregation occurs in polygynous populations that are commonly linked by a dependence on sexual (size) dimorphism (*Bon & Campan, 1996*; *Bowyer, 2004*; *Ruckstuhl & Neuhaus, 2005*). The first, the Predation Risk Hypothesis (PRH), or Reproductive Strategy Hypothesis, postulates that sexual segregation results from each sex investing differentially in strategies to maximise their own reproductive success (*Main & Coblentz, 1996*). Females, perhaps accompanied by dependent and vulnerable young, avoid higher predation risks and forage in places or at times that offer lower nutrient intake (*Corti & Shackleton, 2002*; *Main & Coblentz, 1996*). In contrast, males may opt to maximize their competitive advantage by investing in growth and therefore exploit nutrient rich habitats despite the increased risk of predation (*Clutton-Brock, Guinness & Albon, 1982*; *Prins, 1989*). The second, the Forage Selection Hypothesis (FSH), or Sexual Dimorphism Body-size Hypothesis, predicts that allometric differences in body size, bite size, energy requirements and/or fibre digestion between the sexes can lead to differences in diet selection and habitat choice (*Barboza & Bowyer, 2000*; *Demment, 1982*; *Main & Coblentz, 1996*). Individuals with a smaller gut system are less able to digest lower quality food items and are predicted to feed on higher quality diets than larger conspecifics (*Barboza & Bowyer, 2001*; *Demment & Van Soest, 1985*). The third, the Activity Budget Hypothesis states that body size dimorphism promotes differences in activity budgets and synchrony of these behaviours results in aggregation of the sexes (*Conradt, 1998*; *Ruckstuhl & Neuhaus, 2002*; *Ruckstuhl & Neuhaus, 2000*). Here, energetic requirements and digestive abilities predict that the smaller sex will spend more time foraging and less time resting and digesting than the larger sex (*Ruckstuhl, 1999*; *Ruckstuhl, 1998*), such that females will congregate to engage in foraging together while males rest in areas with low risk of predation. A fourth, and much less studied hypothesis is the Weather Sensitivity Hypothesis which suggests that animals could differ in their weather sensitivity (e.g., *Young & Isbell, 1991*), where larger individuals, often males, suffer higher absolute heat loss and therefore opt for warmer habitats often at the expense of foraging availability (*Conradt et al., 2001*)

There is little consensus as to what are the driving mechanisms underpinning sexual segregation (*Conradt & Roper, 2000*; *Ruckstuhl et al., 2006*). Typically, studies are unable to tease apart which hypothesis best predicts why species segregate, often suggesting that multiple hypotheses could be the cause (*Alves et al., 2013*; *Bonenfant et al., 2007*; *Loe et al., 2006*). One reason for this could be that much of the research concentrates on ungulates (*Alves et al., 2013*; *Bon & Campan, 1996*; *Bowyer & Kie, 2004*), in particular

ruminants (*Bowyer & Kie, 2004*) in systems that are notoriously difficult to study in the wild (*Michelena et al., 2004*). It is not always feasible in such free-ranging mammalian systems to collect the physiological and behavioural data necessary to explicitly separate and test these competing hypotheses. Furthermore, in some study locations, their natural predators have disappeared, rendering it difficult to explore the effects of predation risk. Finally, many species that have been studied are dietary specialists, meaning that variations in diet quality may chiefly depend on differences between particular plants, even those of the same species, and thus dietary intake is difficult to determine accurately without measuring the nutrient quality of each mouthful (*Dove & Mayes, 1996*).

The pheasant, *Phasianus colchicus,* provides a novel, alternative system to ungulates to try to tease apart hypotheses of segregation based on sexual size dimorphism. Many avian species sexually segregate (e.g., northern giant petrels (*Macronectes halli*) (*González-Solís, Croxall & Wood, 2000*) and Great bustards (*Otis tarda*) (*Palacín et al., 2009*), however such segregation is often attributed to differential settlement of the sexes in discrete habitats, often over large distances, and is primarily explained by and manifest in differences in migration behaviours or broad differences in habitat use (*Catry et al., 2006*). Pheasants, however, exhibit a pattern of fine scale sexual segregation similar to the regularly studied ungulate, in that they segregate, outside of the mating season, within the same area as each other (*Hill & Ridley, 1987*; *Whiteside et al., 2018*). Specifically, during the late autumn and winter, females aggregate in same sex groups whereas males avoid both males and females (*Whiteside et al., 2018*). Such segregation persists until early March (*Hill & Ridley, 1987*; *Hill & Robertson, 1988*) when harems of females visit and eventually join territory-holding males, which likely reduces their harassment by other males (*Ridley & Hill, 1987*) and allows females to decrease their vigilance levels and so increase time spent foraging (*Whiteside, Langley & Madden, 2016*), until they independently start to nest and incubate their eggs (*Taber, 1949*). Pheasants become sexually dimorphic by three weeks of age (*Whiteside et al., 2017*), and chicks in captivity (<8 weeks old) exhibit preference for their own sex (*Whiteside et al., 2017*). As adults, males have highly conspicuous plumage and are 40% larger than the cryptic females (*Wittzell, 1991*). Between October and February, before the breeding season, released first year pheasants show increasing levels of segregation (*Whiteside et al., 2018*). Therefore, pheasants present a system that is analogous to the currently studied ungulates, yet this novel taxa may offer more general insights as to how size-dimorphism may influence sexual segregation for species with fine scale sexual segregation.

Mechanisms that drive sexual segregation of pheasants are poorly understood. In the wild, sexual segregation of adults was observed at both a spatial and temporal scale that could not be explained by crude measures of habitat structure, although the homogenous nature of the study and reliance on data from supplementary feeding sites meant that fine scale differences in habitat structure may have not been captured (*Whiteside et al., 2018*). During early life when the sexes differ little in their size, juvenile pheasants reared in captivity in an environment that controlled for habitat selection and diet, albeit under unnaturally high numbers but at a sex ratio analogous to that observed in the wild, exhibited strong preferences for same-sex individuals in binary choices which may drive segregation (*Whiteside et al., 2017*). We suggested that at this stage, females aggregate with

other females in response to male aggression. When adults, body-size dimorphism is more pronounced and the habitat is more heterogeneous and consequently we may find other factors influence sexual segregation. We excluded the possibility that segregation in wild-living pheasants arises because of the Weather Sensitivity Hypothesis. Pheasants typically spend the majority of their time in hedgerows (*Hill & Robertson, 1988*), a place that offers both protection from the weather and a high abundance of natural forage (e.g., insects, leaves, berries, wild fruit and nuts (*Lachlan & Bray, 1973*)). This violates a major assumption of the Weather Sensitivity Hypothesis in that a habitat should offer either shelter or good foraging opportunity, not both. Therefore, in this study we tested the remaining three main hypotheses for sexual segregation that relate to sexual (size) dimorphism in free-living adult pheasants.

Firstly, pheasants are at risk from terrestrial predators, such as the fox, *Vulpes vulpes* (*Hessler et al., 1970*; *Krauss, Graves & Zervanos, 1987*), and aerial predators, such as goshawks, *Accipter gentilis*, sparrow hawks, *Accipiter nicus,* and buzzards, *Buteo buteo,* (*Kenward et al., 2001*; *Kenward, Marcström & Karlbom, 1981*) resulting in high mortality rates of up to 80% in the first month after release into the wild (*Hessler et al., 1970*). Such predation risk is unlikely to explain segregation due to risk aversion by females caring for young because segregation occurs prior to the first breeding season (*Hill & Ridley, 1987*; *Whiteside et al., 2018*). However, outside the breeding season, males may still opt for a riskier foraging and movement strategy if there is a benefit for their growth. Therefore, if the Predation Risk Hypothesis influences segregation then we expect that predation risk (in terms of willingness to approach an area where predators have recently visited) and consequently predation rates will differ between sexes.

Secondly, adult pheasants are dietary generalists (*Hoodless et al., 2001*). Dietary choices can be assayed post-mortem from food that is well preserved and identifiable in the crop of birds that have been shot during recreational hunting (*Whiteside, Sage & Madden, 2015*). Post-mortem analyses of these shot birds also allows for the measurements of gut morphometrics. Both these factors permit explicit testing of the Forage Selection Hypothesis. While gut size is likely to co-vary with body size, with larger males possessing larger guts, allometric differences are meaningful from a nutritional point of view as larger guts are more effective digesters and absorbers of low quality diet (*Barboza & Bowyer, 2000*). In addition, body size dimorphisms may correspond to differences in food processing efficiency via bite size (*Illius & Gordon, 1987*) or grinding capacity in the gizzard (*Putaala & Hissa, 1995*). If the Forage Selection Hypothesis influences segregation then we expect that the larger sex (males in the case of pheasants) will have larger guts and heavier gizzards, and this will be matched by a difference in diet corresponding to spatial or temporal segregation.

Thirdly, differences in behaviour between the sexes in this large, diurnal and conspicuous species have been observed during the breeding season (*Whiteside, Langley & Madden, 2016*), however little is known about behavioural differences outside the breeding season during the period when pheasants sexually segregate. If the Activity Budget Hypothesis operates then we expect that during the periods of sexual segregation male and female pheasants will differ in their behaviour, specifically state behaviours known to influence

**Table 1   Representing a description, for each year, of the conditions each bird was reared under, the numbers per house, the release sex ratio along with what data was collected.** The default environment was analogous to current industrial rearing conditions and acted as our control; a barren, spatially simple environment that offered a monotonous chick crumb diet that as *ad lib* and in excess. Within the parenthesis next to the measures denotes the sample size and which hypotheses it was used to test.

| Year | Rearing (day 1—release day) | Release day (day 43–62) | Post release (release day until 1 March) | Shooting season (1 October–1 February) |
|---|---|---|---|---|
| *2012* | 10 replicates of three dietary treatments (1) 1% mealworms; (2) 5% mixed seed; (3) Control Rearing numbers = 30 per house Release sex ratio(f:m) = 50:50 Large release number (game keeper) | Mass (871) Tarsus (871) | Foraging behaviour (167: ABH) Mortality (13: PRH) | Mass (233) Tarsus (233) Gut morphology (129: FSH) Crop mass (159: FSH) |
| *2013* | 3 × 2 design. 10 replicates of 3 dietary treatments (1) 1% mealworms; (2) 5% mixed seed; (3) Control 15 replicates of 2 structural treatments (1) access to perches; and (2) control (no perches) Rearing numbers = 30 per house Release sex ratio(f:m) = 46:54 Large release number (game keeper) | Mass (901) Tarsus (901) | Foraging behaviour (214: ABH) Vigilance behaviour (214: ABH) Resting behaviour (214: ABH) Walking behaviour (214: ABH) Mortality (18: PRH) | Mass (202) Tarsus (202) Crop samples (147: FSH) |
| *2014* | No environmental treatments. All had control environment with the addition of Supplementary mealworms and mixed seed and had access to perches Rearing numbers = 50 per house Release sex ratio(f:m) = 46:54 Small release number (no game keeper) | None | Feeder use relative to possible predation events (50: PRH) | None |
| *2015* | No environmental treatments. All had control environment with the addition of Supplementary mealworms and mixed seed and had access to perches Rearing numbers = 50 per house Release sex ratio(f:m) = 46:54 Small release number (no game keeper) | Mass (194) Tarsus (194) | Mortality (42: PRH) | None |

**Notes.**

FSH, Forage Selection Hypothesis; PRH, Predation Risk Hypothesis; ABH, Activity Budget Hypothesis.

sexual segregation in ungulates (e.g., foraging, locomotion or resting time (*Ruckstuhl, 1998*)). Male pheasant chicks in captivity were more aggressive than females and this aggression provides potential mechanisms driving segregation (*Whiteside et al., 2017*). However, pheasants in the wild exhibit little aggression during the periods that they sexually segregate, with male-male aggression rising from the start of the breeding season and peaking mid-breeding season (*Ridley, 1983*).

To tease apart these hypotheses we draw on two populations of individually identifiable pheasants that were reared in captivity, measured and then released into the wild (see Table 1 for a description of each population and what was measured). The first population was released into an environment that did not have predator control or recreational shooting. Although initial release density was unnaturally high (∼200 birds in a 0.5 Ha

release pen), after one month birds had dispersed across the study site. At this point the population density was ~40 birds/km$^2$ which matched those of wild populations, falling within the density (16–54 birds/km$^2$) for wild pheasants living in managed farmland in Austria and the density (0.6–64 birds/km$^2$) for pheasants in their native range in China (Li 1996 in *Johnsgard, 1999*). Crucially, within one month of release this population showed clear sexual segregation which became pronounced as the year progressed (see *Whiteside et al., 2018*). Releasing on sites without either predator control or recreational hunting allowed us to measure natural predation rates. By using a system of motion sensitive camera traps at feeding sites we were able to determine: (1) if pheasants avoid areas where foxes had been present: and (2) if sexes differ in their willingness to enter an area where a fox had previously been seen (essential for the PRH). The second population were birds that were reared in captivity and released into the wild in large numbers (~350 birds in a 0.5 Ha release pen) as part of a restocking programme for commercial shooting. However, these birds were released on a much larger site and within a few weeks of birds dispersing resulted in much lower density over the entire estate. On this site there was managed predator control and the birds were subject to hunting. Birds released using this method still show patterns of sexual segregation (*Hill & Ridley, 1987*) similar to that observed in pheasants released at lower density. Releasing onto a site that has recreational hunting allows for us to conduct post mortem analysis that: (1) allows us to determine the extent of sexual dimorphism in body size (essential for PRH, FSH and ABH) and in gut morphology (essential for FSH); and (2) acts as a dietary snapshot, whereby crop sample analyses allows us to determine diet (essential for FSH). Observing behaviour of pheasants during this period allows us to determine activity budgets (essential for ABH).

## MATERIAL AND METHODS

### Rearing

In May 2012 and 2013 we reared 1,800 pheasants (900 / year) from one day old as part of a long-term study to determine how early rearing conditions can influence development and post release mortality. Chicks were placed in houses of 30 individuals with an equal sex ratio and each house was randomly allocated a rearing treatment. While not relevant to this study, the treatments included differences in supplemented diet in 2012 (see Table 1 and *Whiteside, Sage & Madden, 2015*) and access to perches in 2013 (see Table 1 and *Whiteside, Sage & Madden, 2016*) as well as controls. In 2014 and 2015 we reared a further 400 pheasants (200 / year) from one day old and housed them in groups of 50 under identical conditions (see Table 1). Each year the birds were housed for two weeks in heated sheds (2012/13: 1.3 m × 1.3 m; 2014/15: 2 m × 2 m) and were then given access to an additional open grass run (2012/13 = 1.3 m × 6.8 m; 2014/15: 4 m × 12 m) until release. All chicks were provided with age specific commercial chick crumbs (Sportsman Game Feeds) *ad lib* and in excess. Water was provided *ad lib*. In all four years, birds were marked with patagial wing tags (Roxan Ltd, Selkirk, UK) for identification with additional white PVC wing tags (25 mm × 75 mm) with individually unique identifying numbers which could be viewed from several tens of meters away.

### Release into the wild

Following rearing, birds were randomly mixed from across different housing groups and placed into open topped pens. Release pens typically consisted of wire mesh fences $\sim2$ m high enclosing an extensive area of woodland (*GWCT, 1991*). In these pens birds were provided with food and water *ad lib*. Birds could disperse from these pens and were free to roam and mix with other released, as well as resident, pheasants. In 2012 and 2013 when birds were approximately seven weeks old we released them onto the Middleton Estate, Hampshire, UK (51°18′N, 1°4′W). The estate, predominantly arable, hosts a game shoot and employs two game keepers to manage the released pheasants through habitat management, providing supplementary food, and controlling predator numbers. Between October and February birds were shot as part of a recreational shoot. In 2014 and 2015 when birds were ten weeks old they were released at North Wyke Farm, Devon (50°77′N, 3°9′W). This site is grazed by cattle and sheep and no game shooting or predator control occurred there. Forty feeders, filled with wheat, were placed within the pen ($n = 4$) and in the surrounding countryside ($n = 36$) at a density of 0.16 per hectare. In 2014, each feeder was continuously monitored with Bushnell® Trophy motion activated cameras. All animals that visited a feeder and its surrounding area were photographed and the images were then viewed manually to record the time that pheasants and foxes visited the feeder site. Individual pheasants could be identified from their wing tag numbers.

### Body size dimorphism

We recorded the mass (Slater Super Samson spring balance—precision 5 g) and tarsus length (precision 1 mm) of all birds upon release into the wild and for birds released in 2012 and 2013 we scored the same measures within four hours of them being shot which occurred four to seven months after their release.

For the released populations we used a General Linear Mixed Model (GLMM) to identify whether males differed from females in their mass and tarsus length with rearing treatment and sex as fixed factors and the rearing house as a random factor, with all two way interactions included (Table 2). In 2014 and 2015 all birds were reared under identical conditions and therefore rearing treatment was not included in the model. For birds shot in 2012 and 2013 a General Linear Model (GLM) was used to ask if sexes differed in mass and tarsus length as adults. The bird's age when shot, its rearing treatment and all two way interactions were included in the GLM (Table 2).

### A test for the Predation Risk Hypothesis: first appearance at a feeder after the presence of a fox

We recorded every sighting of a fox and pheasant at each of our feeder sites during December 2014 and January 2015 using the motion camera traps. In order to test whether the presence of foxes at feeders was an indicator of a risky environment that pheasants attended to, we tested whether birds took longer to return to feeders after a fox had been there compared to the time it took them to appear at a feeder after a time-matched control point the previous day. We excluded instances where there were low visitation rates at a feeder, indicated by long periods (>420 mins) between the time-matched control point and

**Table 2  The distribution, response variables, explanatory variables and random factors for all GLM and GLMMs used in the study.**

| Question | Distribution | Response | Explanatory factors | Random factors |
|---|---|---|---|---|
| *Do sexes differ in mass upon release into the wild?* | Normal | Mass | Sex of focal Rearing treatment (2012 and 2013 only) | House |
| *Do sexes differ in tarsus length upon release into the wild?* | Normal | Tarsus length | Sex of focal Rearing treatment (2012 and 2013 only) | House |
| *Do sexes differ in mass when shot as an adult?* | Normal | Mass | Sex of focal Age when shot Rearing treatment | |
| *Do sexes differ in tarsus length when shot as an adult?* | Normal | Tarsus length | Sex of focal Age when shot Rearing treatment | |
| *Do sexes differ in their gut morphology when shot as adults* | Normal | Length/Mass/Volume | Sex of focal Age when shot Rearing treatment | |
| *Do sexes differ in the mass of food found in their crop?* | Normal | Mass | Sex of focal Age when shot | |
| *Do sexes differ in the time spent foraging (2012)?* | Normal | Percentage of time spent foraging (logit transformed) | Sex of focal Time of day (am/pm) Rearing treatment Degree of aerial protection (Open/Closed canopy) | |
| *Do sexes differ in foraging, walking and resting behaviours (2013)?* | Binomial | Likelihood performing behaviour | Sex of focal Time of day (am/pm) Rearing treatment Degree of aerial protection (Open/Closed canopy) | |

the first pheasant appearing. This cut-off point was meaningful and discrete (Fig. 1). This left a subset of the previous data including 110 cases. We used a *t*-test to ask if pheasants took longer to approach a feeder if it had been visited by a fox compared a time matched control point the previous day. We then looked at each appearance of a fox and recorded the sex of the next pheasant to enter the same feeder within a subsequent 30 min. We used a binomial test to determine if sexes differed in their likelihood of approaching a feeder following a fox visit.

## A test for the Predation Risk Hypothesis: do predation rates differ with sex

In 2012 and 2013 we conducted searches of areas surrounding the release site to retrieve birds that had been killed by predators. Searches were conducted daily from August-October and then again in February. During the hunting season (late October to February) the area was visited less frequently but more methodically by beaters who were engaged in driving the game to the waiting hunters. They were informed of the project and searched for carcasses and tags as they walked through the site. In 2015 we collected birds that had been killed by a predator by searching the release site and surrounding areas for carcasses, locating these either directly or guided by radio tags placed on 50 birds. In 2014 we did not conduct detailed searches for carcasses. A binomial test was used to test whether predation numbers differed between sexes with the expected outcome based on the released sex ratio.

## A test for Forage Selection Hypothesis: measuring gut morphology

We collected linear gut measures (oesophagus, intestine, colon and ceca) and gut masses (oesophagus, intestine, crop, gizzard and ceca; for methods see *Leopold, 1953*) of 186 birds

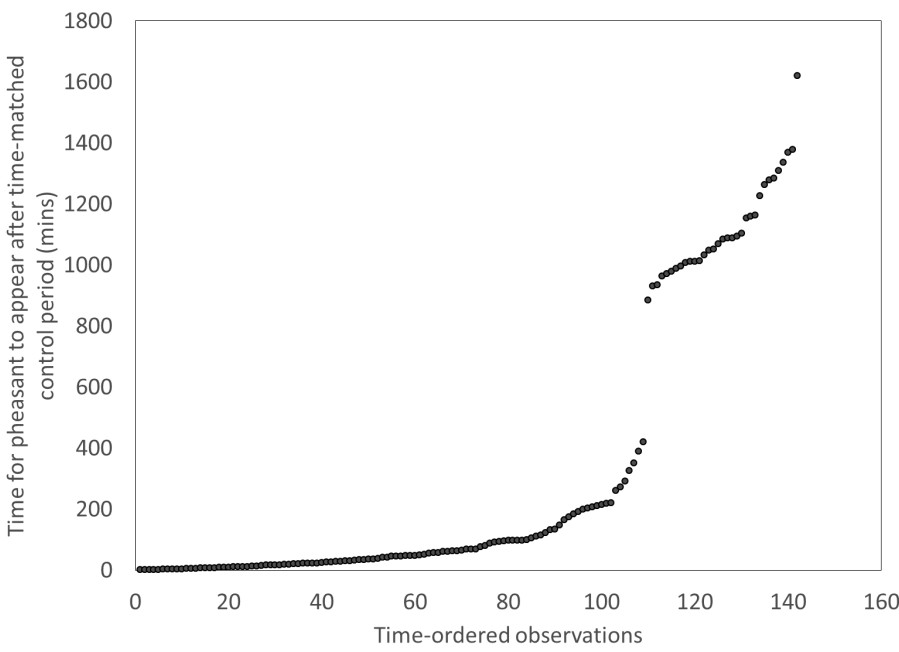

**Figure 1 Distributions of the times for pheasants to appear after a time-matched control period, showing a clear break after ∼420 mins**

shot in 2012. Ceca length and ceca mass were calculated as the average for both ceca. Gizzard volume was measured as the height multiplied by width. We used a GLM to test whether gut morphologies differed with sex. Since rearing treatment can influence gut morphology (*Whiteside, Sage & Madden, 2015*) we included the rearing treatment and the age at which the bird was shot as fixed factors (Table 2).

## A test for Forage Selection Hypothesis: measuring crop contents in the wild

In 2012, we removed the crops of 159 shot birds and measured the mass before and after removal of its contents. Measures were transformed using the $\sqrt{(x)}$ transformation to meet assumptions of normality and then a GLM was used to test if mass of crop content differed between males and females. In 2013, we emptied the crops of 168 shot birds and quantified their contents. We used a GLM to ask whether males differed from females in the number of different food items discovered in their crops. Diet availability will change depending on when the bird is shot and diet choice is influenced by rearing treatment (*Whiteside, Sage & Madden, 2015*) therefore in both GLMs we included rearing treatment and the age of the bird when it was shot in the model (Table 2). We separated all known items into common food categories (Wheat, Maize, Grass, Oil Seed Rape, Insects, Seeds, Galls, Acorns) and used binomial tests to determine if sexes differed in the likelihood of their crop containing food of each category.
### A test for Activity Budget Hypothesis: measuring behaviour in the wild

In 2012, we conducted continuous focal follows, for a maximum of 10 min, observing 167 released pheasants between 18 September and 5 November. All observations were conducted from inside a vehicle at a distance of >10 m (e.g. *Whiteside, Langley & Madden, 2016*) either during the first 2 h after first light or the final 2 h before last light (*Taber, 1949*). We recorded the total time we observed the birds, the time spent foraging and the number of foraging bouts they performed. A foraging bout began with the lowering of the head and neck towards the ground and ended when the neck was raised. The proportion of time an animal spent foraging was normalised using a logit transformation: $\log(y/1-y)$ (*Warton & Hui, 2010*) and a GLM was used to test if the percentage of time spent foraging differed between the sexes.

In 2013, between 15 August and 16 September we conducted a series of 10 min focal follows with an instantaneous point sampling procedure at 30 s intervals on 214 pheasants. This allowed us to collect state behaviours; in particular time spent performing resting, foraging and locomotive behaviours and also aggressive interactions. All birds were identifiable from their wing tags and were observed from a distance so as to not disturb their natural behaviours. Vigilance was described as sitting or standing with neck extended and eyes open. Resting was determined as either standing or lying with eyes closed. Aggressive interactions included threats, run threats, lateral struts and fights (see *Hill & Robertson, 1988*; *Ridley, 1983*). We used a Generalised Linear Model with a binomial distribution and a probit link function to ask if sexes differed in their vigilance, walking and resting likelihoods. In all models early rearing environment, time of day (AM or PM), and degree of aerial protection (open or closed) were included as fixed factors, and all two-way interactions were assessed (Table 2).

### Statistical analyses

All GLM and GLMM analyses were conducted using SPSS v23 (Chicago, IL, USA). All models were visually inspected for homogeneity of variance, normality of error and linearity.

### Ethical statement

All birds were reared using commercial procedures that adhere to the DEFRA Code of Practice for the Welfare of Game Birds Reared for Sporting Purposes (*DEFRA, 2009*). During rearing, minimal handling was used for obtaining morphometrics and placing birds in testing chambers. In 2012 and 2013, once birds dispersed from the release pen, gamekeepers supplied supplementary feed and water, which was reduced after the shooting season (from 1st February). The birds were shot as a part of a commercial shoot, and were not specifically shot for this study. In 2014 and 2015 released birds were attended to by the authors and there was no shooting on the study site. The work was approved by the University of Exeter Psychology Ethics Committee and conducted under Home Office licences number PPL 30/3204 & PPL 30/2942

**Table 3** The mean mass in grams (range) of males and females released into the wild for three rearing seasons and the mass of adult birds shot in 2012 and 2013.

| Year (age) | Male mass (g) | Female mass (g) | df (df-error) | F | P |
|---|---|---|---|---|---|
| 2012 (50 days) | 643.69 (400–800) | 536.32 (300–760) | 1 (865) | 688.29 | <0.001 |
| 2013 (43 days) | 489.42 (350–630) | 412.07 (220–540) | 1 (889) | 169.90 | <0.001 |
| 2015 (62 days) | 738.31 (556–936) | 607.82 (466–726) | 1 (192) | 233.37 | <0.001 |
| 2012 (Adult) | 1,577.18 (1,140–2,200) | 1,220.32 (1,010–1,510) | 1 (124) | 830.03 | <0.001 |
| 2013 (Adult) | 1,574.63 (1,170–2,000) | 1,223.66 (920–1,500) | 1 (223) | 503.33 | <0.001 |

**Table 4** The mean tarsus length in mm (range) of males and females released into the wild for three rearing seasons and the tarsus length of adult birds shot in 2012 and 2013

| Year (age) | Male tarsus (mm) | Female tarsus (mm) | df (df-error) | F | P |
|---|---|---|---|---|---|
| 2012 (50 days) | 69.01 (55–79.2) | 63.28 (52.7–78) | 1 (865) | 144.09 | <0.001 |
| 2013 (43 days) | 62.70 (51.2–69.9) | 58.56 (45.7–65) | 1 (873) | 392.11 | <0.001 |
| 2015 (62 days) | 72.57 (67.2–81.1) | 65.48 (67.2–81.1) | 1 (192) | 312.50 | <0.001 |
| 2012 (Adult) | 79.97 (72.95–88.2) | 70.41 (63.2–76.15) | 1 (221) | 590.53 | <0.001 |
| 2013 (Adult) | 80.79 (75.25–89.4) | 71.34 (61.05–78.1) | 1 (186) | 547.3 | <0.001 |

## RESULTS

### Are birds sexually dimorphic during the period of the study?

Males were significantly heavier (Table 3) and had longer tarsi (Table 4) than females upon release into the wild and when they were shot as adults prior to their first breeding season

### A test for the Predation Risk Hypothesis: do sexes differ in their likelihood of being the first to approach a feeder visited by a fox?

Pheasants took significantly longer (approx. 2.5 times longer) to appear at a feeder after a fox was present ($193 \pm 35$ mins) than after a time-matched control point the previous day ($76 \pm 9$ mins) ($t_{109} = 3.37$, $P = 0.001$, Fig. 2). There was no difference in the likelihood that a male or female would be first to approach a feeding site in the following 30 min after the sighting of a fox (female = 20; male = 30; Binomial test: $P = 0.20$).

### A test for Predation Risk Hypothesis: do predation rates differ with sex?

Predation did not differ with sex in the first 8 months after release in 2012 (Binomial tests: 2012: female = 8; male = 5, $P = 0.58$); 2013 (Binomial test: female = 6; male = 12: $P = 0.20$); and 2015 (Binomial test: female = 15; male = 27: $P = 0.12$).

### A test for Forage Selection Hypothesis: do sexes differ in their gut morphology?

Males significantly differed from females in all aspects of measured gut morphologies with larger and heavier gut regions, crops and gizzards (Table 5).
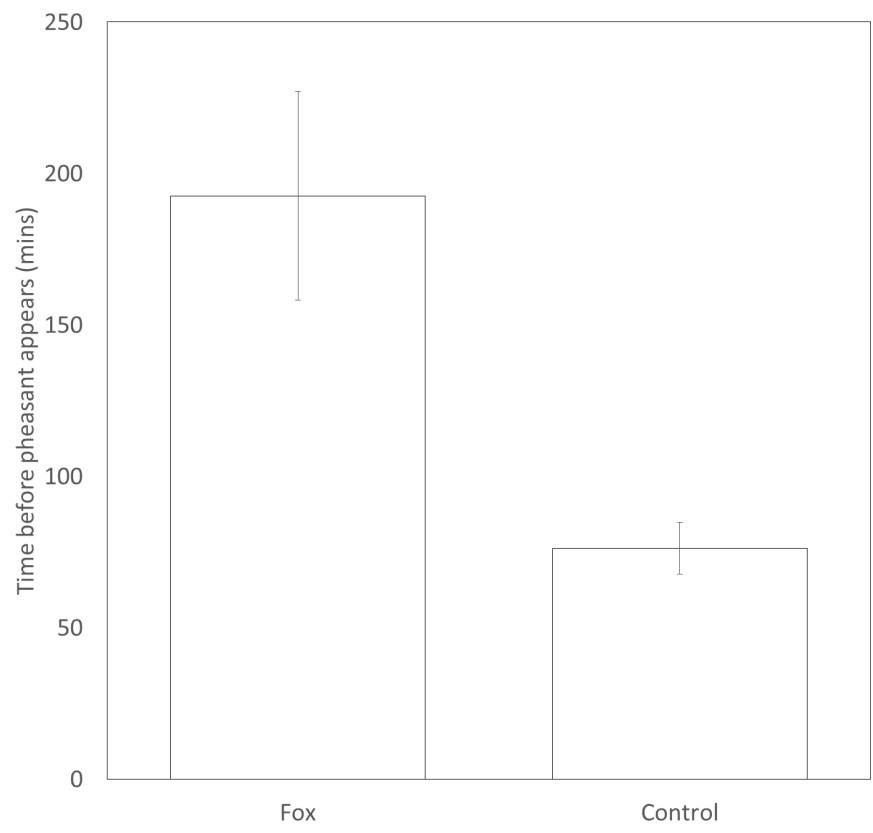

**Figure 2** **Time delay before a pheasant appears at a feeder after a fox has been present or a paired, time-matched control period 24 hrs before the fox was sighted. Error bars = ±1SE**

## A test for forage selection hypothesis: does diet composition differ between sexes?

Males (17.38 g ± 1.62) shot in 2012 had significantly more forage in their crops compared to females (12.75 g ± 1.08) ($F_{1,131} = 4.29$, $P = 0.040$, Fig. 3). However, males (3.33 ± 0.16) did not differ from females (3.52 ± 0.27) in the variety of food items found within their crop ($F_{1,147} = 0.93$, $P = 0.34$). The sexes appeared to utilise a similar diet with both sexes carrying similar proportions of eight common food types in their crops. Males did not differ from females in the likelihood that their crop would contain wheat, maize, grass, oil seed rape, insects, seeds, galls or acorns (Table 6).

## A test for Activity Budget Hypothesis: do other behaviour differ between sexes?

Sexes did not differ in their percentage of time spent foraging; the length of each foraging bout (Table 7); their likelihood of being vigilant ($F_{1,186} = 0.20$, $P = 0.66$) or their likelihood of walking ($F_{1,186} = 2.54$, $P = 0.13$). There was only one incidence of resting behaviour during the focal watches which was demonstrated by a male and there were no aggressive interactions performed hence sex differences in these behaviours could not be compared.

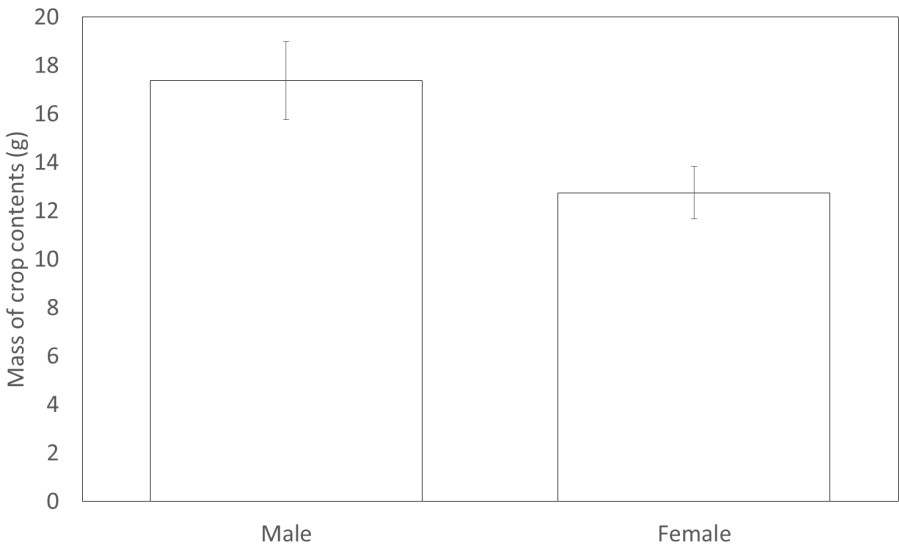

**Figure 3** The mean mass of crop contents from birds shot in 2012. Error bars indicate ±1 SE

**Table 5** Mass (g) and lengths (mm) of male and female gut morphologies (degrees of freedom = 1,128) for birds shot as adults in 2012.

| Dependent variable | Sex | Mean | Std. error | Relative difference | F | P |
|---|---|---|---|---|---|---|
| Oesophagus Length | Female | 111.75 | 1.82 | 1.10 | 23.720 | <0.001 |
| | Male | 123.00 | 1.43 | | | |
| Oesophagus Mass | Female | 1.79 | 0.08 | 1.27 | 23.227 | <0.001 |
| | Male | 2.28 | 0.06 | | | |
| Crop Mass | Female | 3.67 | 0.16 | 1.22 | 16.318 | <0.001 |
| | Male | 4.48 | 0.12 | | | |
| Gizzard Mass | Female | 22.55 | 0.53 | 1.18 | 33.098 | <0.001 |
| | Male | 26.45 | 0.42 | | | |
| Intestine Length | Female | 1,146.23 | 15.78 | 1.09 | 25.942 | <0.001 |
| | Male | 1,248.35 | 12.37 | | | |
| Intestines Mass | Female | 15.73 | 0.41 | 1.17 | 25.475 | <0.001 |
| | Male | 18.34 | 0.32 | | | |
| Colon Length | Female | 100.31 | 2.02 | 1.08 | 11.128 | <0.001 |
| | Male | 108.88 | 1.58 | | | |
| Colon Mass | Female | 2.08 | 0.09 | 1.27 | 24.731 | <0.001 |
| | Male | 2.65 | 0.07 | | | |
| Average Ceca Length | Female | 214.67 | 3.90 | 1.12 | 26.408 | <0.001 |
| | Male | 240.13 | 3.06 | | | |
| Average Ceca Mass | Female | 3.33 | 0.11 | 1.21 | 26.301 | <0.001 |
| | Male | 4.04 | 0.09 | | | |
| Gizzard Volume | Female | 45,070.23 | 1,501.27 | 1.13 | 9.514 | <0.001 |
| | Male | 50,955.55 | 1,177.55 | | | |

**Table 6** The percentage of males and females in the population with crop contents containing certain food for birds shot in 2013 with associated binomial statistics.

| Food item | Males (%) | Females (%) | P |
|---|---|---|---|
| Wheat | 62.39 | 74.51 | 0.19 |
| Maize | 58.12 | 56.86 | 0.54 |
| Grass | 68.38 | 62.75 | 0.42 |
| Oil Seed Rape | 8.55 | 7.84 | 0.58 |
| Insects | 10.26 | 13.73 | 0.33 |
| Seeds | 47.86 | 56.86 | 0.24 |
| Galls | 7.69 | 7.84 | 0.58 |
| Acorns | 23.93 | 11.76 | 0.08 |

**Table 7** The mean percentage of time spent foraging and the mean foraging bout length for males and females after release into the wild.

| Behaviour | Year | Male | | Female | | df (df-error) | F | P |
|---|---|---|---|---|---|---|---|---|
| | | Mean | SEM | Mean | SEM | | | |
| Percentage foraging | 2012 | 32.41 | 2.24 | 35.23 | 3.2 | 1(139) | 0.01 | 0.98 |
| | 2013 | 33.1 | 2.1 | 29.53 | 2.19 | 1(186) | 1.84 | 0.18 |
| Forage bout length | 2012 | 14.63 | 1.29 | 16.09 | 1.49 | 1(139) | 0.33 | 0.57 |
| | 2013 | 10.44 | 0.66 | 9.79 | 0.69 | 1(160) | 0.46 | 0.5 |

## DISCUSSION

During the late autumn and winter, when pheasants show an increased tendency to sexually segregate (*Whiteside et al., 2018*), pheasants exhibited strong sexual size dimorphism. Adult males were significantly heavier than females and had larger gut dynamics and heavier gizzards, however, we found little evidence that sexes differed in behavioural and dietary measures that are predicted to explain sexual segregation that we observed according to the three existing hypotheses purporting to explain segregation due to size dimorphism (*Bleich, Bowyer & Wehausen, 1997*; *Bonenfant et al., 2007*; *Conradt, 1998*; *Ruckstuhl, 1998*).

Males pheasants were 1.3× heavier than females which is similar to levels of dimorphism seen in Northern giant petrel (*Macronectes halli*) (1.25: *González-Solís, 2004*) but less than the great bustard (*Otis tarda*) (2.48: *Alonso et al., 2009*), two avian species that are known to sexually segregate. The extent of pheasant size dimorphism also closely matched that observed in mammals exhibiting sexual segregation (red deer, *Cervus elephus*, (1.33: *Clutton-Brock, Guinness & Albon, 1982*; *Weckerly, 1998*), merino sheep, *Ovis aries*, (1.50: *Michelena et al., 2004*), bighorn sheep, *Ovis canadensis*, (1.43: *Blood, Flook & Wishart, 1970*). Pheasants share similarities with such ungulates being highly terrestrial and sedentary, in contrast to the wide ranging and migratory petrel and bustard.

Pheasants may reduce their risk of predation by avoiding feeders recently visited by foxes. This was indicated by their increasing the lag between feeder visitations after a fox visit compared to a time-matched control point the previous day, suggesting that time taken to visit a feeder after a fox has been there provided a good assay of risk sensitivity. However, we found that sexes did not differ in their exposure to fox predation risk, at

least at feeders, indicated by their likelihood of being the first bird to visit a feeding site after a fox had been present. Such patterns of risk sensitivity may differ at non-feeding sites. A better understanding of concurrent predator and pheasant movement would also help clarify how predator risk might influence segregation. This similarity between the sexes in their willingness to enter an area previously visited by a fox, the most common predator of pheasants in the UK, may explain why we also found no significant differences in predation rates between sexes across all three years in which we monitored predation. This may appear somewhat surprising, particularly if the size and colouration of males makes them more conspicuous and indeed we tended to find more males predated than females, perhaps because we used visual searches favouring detection of the larger and more visible males. However, other studies using radio collared release populations, also revealed no differences in predation rates between sexes (*Musil & Connelly, 2009*; *Turner, 2007*). In contrast to the breeding season when nesting females are more exposed to foxes either while sitting on the nest or attending dependent young, pheasants of both sexes faced similar predation risks and suffered similar predation rates during autumn/winter prior to the breeding season. During this period, females may not discriminate against high predation risk habitats because they do not have to protect their young. In contrast, immediately after the breeding season, precocial pheasant chicks frequently stay with their mother for over 28 days (*Riley et al., 1998*) and during this time (June-August) females with young may occupy low risk habitats. However, our recording periods did not encompass this time and our females were all birds hatched in the spring and therefore had not yet bred. The similarity between the sexes in predation risk and consequent predation rates suggests that the Predation Risk Hypothesis may not adequately account for sexual segregation in pheasants outside the breeding season.

During the period that pheasants sexually segregate, females were smaller in both body size and in all aspects of their gut morphology. The longer intestinal system in male pheasants suggests a better ability to efficiently digest a lower quality diet (*Moss, 1983*). Larger gizzards in the males also suggest a more effective grinding mechanism, perhaps allowing the digestion of harder food items (*Putaala & Hissa, 1995*). Such gastro-intestinal dimorphism is common in sexually segregating ungulates, often with males having a larger rumen, small intestine and colon, allowing for them to forage on much less digestible forage (*Barboza & Bowyer, 2000*). Such differences in morphologies could cause sexes to differ in their diet, perhaps with female pheasants choosing higher quality foods which are easy to grind in the gizzard. However, we did not observe differences in crop content when considering a suite of common food items. Similarity in dietary preference was also observed in pheasants when tested as chicks in captivity and presented with a choice of a variety of natural and man-made food items (*Whiteside et al., 2017*). Dietary difference between sexes often occurs, or become more pronounced, in periods leading up to nesting and incubation (*Lewis et al., 2002*; *Nisbet, 1997*). During the same period that data from this study was collected, the degree of sexual segregation becomes stronger as the pheasants get older (*Whiteside et al., 2018*). This might indicate that females begin to differentiate their foraging behaviours more in the run up to the start of the breeding season in March. Our sampling of crop contents finished at the start of February corresponding with the

end of the shooting season, so we may have missed this dietary switch at the advent of the breeding season. Although dimorphic in gastro-intestinal morphology, our findings that sexes do not differ in dietary breadth or composition suggest that the Forage Selection Hypothesis may not adequately explain pheasant sexual segregation witnessed during the pre-breeding period.

Nutrient intake requirements are proportional to body size for many species (*Demment & Van Soest, 1985*), yet we found the dietary breadth and composition was similar for both sexes of pheasants (see above). Therefore, we predicted that males would forage more than females, while exhibiting correspondingly lower levels of alternative behaviours such as walking or vigilance, leading to segregation. However we found that males and female pheasants did not differ in their proportion of time spent foraging. One explanation is that there is a sex difference in foraging efficiency and that males can consume enough nutrients in a similar time period. Such an explanation is supported by the observation that male pheasant chicks were twice as quick as females when presented with a novel food handling challenge (*Whiteside et al., 2017*). A lack of sex differences in foraging has also been observed in desert big horn sheep, *Ovis canadensis mexicana,* (*Mooring et al., 2003*) and musk ox, *Ovibos moschatus* (*Côté, Schaefer & Messier, 1997*). In species where sex differences in time spent foraging are observed it is often attributed to their investment in reproduction (*Lewis et al., 2002*) and differences in parental care roles (*Gray & Hamer, 2001*; *Thaxter et al., 2009*). Although female pheasants will forage more than males prior to nesting (*Ridley & Hill, 1987*; *Whiteside, Langley & Madden, 2016*), this occurs in early spring after hens have abandoned their segregated winter aggregations and joined harems.

Much of the post-ingestion processing of foods in pheasants is conducted in the gizzard (*Putaala & Hissa, 1995*), with coarse material being fermented in the ceca, both of which are smaller in females. Given the consistency in diet between sexes (see above) we predict that females spend longer processing food and thus may be observed resting for longer. However, unlike in some sexually dimorphic ruminants (*Ruckstuhl & Neuhaus, 2002*) we found that males did not differ from females in the likelihood that they would be resting, hence suggesting that resting activity cannot explain patterns of sexual segregation.

The Activity Budget Hypothesis is not restricted to behaviours related to food processing, as synchrony of other behaviours can also result in aggregation of the same sex. For instance, increased movement rates by females have been suggested as a reason for sexual segregation in big horn sheep (*Ruckstuhl, 1998*). However, we found that male pheasants did not differ from females in their occurrence of walking. Likewise, similar movement patterns across the sexes were observed in merino sheep (*Michelena et al., 2004*). All locomotor behaviours that we measured were consistent across sexes, therefore we believe that the Activity Budget Hypothesis fails to predict sexual segregation in pheasants.

Aggression in adult pheasants is rarely seen between the sexes and male-male aggression is typically restricted to the breeding season, peaking at the end of March (*Ridley, 1983*) and therefore it is not surprising that we observed no aggressive interactions at any of our point samples. Consequently, we can, tentatively, reject aggression as a potential behavioural mechanism driving sexual segregation. This contrasts with what we observed in young pheasants in captivity where high levels of aggression was observed in males

(*Whiteside et al., 2017*), and perhaps a memory of these effects can persist over the following months driving segregation post-release. However, this does not explain why the level of segregation becomes stronger over time. Our use of state behaviour recording in this study may have missed sporadic but important aggressive interactions, so if we suspect that segregation is driven by one sex trying to avoid aggression by the other a more detailed recording of aggression is required.

It is important to acknowledge the limitation of using captive-reared pheasants as a model system to help understand sexual segregation. Firstly, they are reared in an unnatural environment. Whilst an artificial rearing environment allows us to control for important factors like experience (e.g., of diet, predators) it may also distort future social dynamics due to the lack of adult role models and a relatively limited and barren physical environment. Secondly, the pheasants are released at high densities, initially much greater than that of natural populations, and this could influence availability of habitat or forage. However, both wild and artificially reared pheasants exhibit sexual segregation post release (*Hill & Ridley, 1987*; *Ridley, 1983*; *Whiteside et al., 2018*), although we cannot confirm that the drivers of segregation match those of truly wild birds. Confirmation would require a similarly detailed study on a wild population where predators could be observed and gut morphology and contents recorded.

Sexual dimorphism is fundamental to a suite of hypotheses that predict why many species, mainly ungulates, sexually segregate at a fine scale (*Pérez-Barbería & Gordon, 1999*). However, although size (and plumage) dimorphism is also observed and pronounced in pheasants, we found that the sexes did not differ in their predation risk, diet or behaviour. We therefore found no support that the Predation Risk, Forage Selection or Activity Budget hypotheses adequately explain sexual segregation of pheasants outside the breeding season. We can conceive of four explanations for this. First, birds (including pheasants) exhibit highly plastic gastro-intestinal systems in order to reduce excess mass which is costly to flight (*Dudley & Vermeij, 1992*; *Gasaway, 1976*; *Whiteside, Sage & Madden, 2015*). This flexibility may mean that gut size, and hence the efficiency of nutrient absorption which underpins the hypotheses we tested, is not as closely linked to body size as in ungulates. Consequently, the differences we observe in body size in pheasants do not match differences in dietary needs or foraging patterns and so do not lead to a segregation of the sexes. Second, segregation may be driven by other inherent, social, factors. Pheasants younger than those tested in this paper prefer to associate with others of the same sex in binary choice tests, perhaps because males are aggressive and so seek out same sex partners to spar with, or because females actively avoid males that may injure them (*Whiteside et al., 2017*). Such preferences developed early in life may persist into early adulthood, even though the adults look markedly different from the appearance of the chicks that we tested. Female pheasants older than those tested in this paper also prefer to associate other females in binary choice tests (*Madden & Whiteside, 2013*). We have not explicitly tested the social preferences of pheasants at the ages we studied in this current paper and we advise this for future work. Alternatively, sexes may socially segregate because they are attracted to each other (Social Attraction Hypothesis), in order to facilitate social learning or to become less conspicuous to predators (*Croft et al., 2003*; *Lingle, 2001*). Further work is required to explore what, if

any, benefits pheasants gain from same-sex social partners. Third, segregation may be due to an attempt to reduce risks of infection by females because sexually dimorphic males can carry higher parasite burdens and avoiding them may lower the risk of transmission (*Ferrari et al., 2010*; *Perkins et al., 2003*). We did not measure parasite loads of pheasants. Finally, although we explicitly tested the hypotheses in isolation, some or all of them may influence or reinforce segregation in conjunction with one another. For example, subtle initial differences in diet preferences may initially lead to same sex aggregations (Forage Selection Hypothesis) in which the sexes learn improved foraging techniques from one another (Social Attraction Hypothesis), which leads to synchronised exploitation of specific food types (Activity Budget Hypothesis). Understanding why the sexes segregate may require a more nuanced and integrative consideration of the current hypotheses.

### Funding
The work was jointly funded by the University of Exeter, the Game and Wildlife Conservation Trust and an ERC Consolidator Award (616474) awarded to Joah R. Madden. The funders had no role in study design, data collection and analysis, decision to publish, or preparation of the manuscript.

### Grant Disclosures
The following grant information was disclosed by the authors:
University of Exeter.
Wildlife Conservation Trust.
ERC Consolidator Award: 616474.

### Competing Interests
The authors declare there are no competing interests.

### Author Contributions
- Mark A. Whiteside and Joah R. Madden conceived and designed the experiments, performed the experiments, analyzed the data, contributed reagents/materials/analysis tools, prepared figures and/or tables, authored or reviewed drafts of the paper, approved the final draft.
- Jayden O. van Horik, Ellis J.G. Langley and Christine E. Beardsworth performed the experiments, authored or reviewed drafts of the paper, approved the final draft.

### Animal Ethics
The following information was supplied relating to ethical approvals (i.e., approving body and any reference numbers):

The work was approved by the University of Exeter Psychology Ethics Committee and conducted under Home Office license numbers PPL 30/3204 & PPL 30/2942.

## Data Availability

Open Research Exeter (ORE): https://doi.org/10.24378/exe.683.

## Supplemental Information

Supplemental information for this article can be found online at http://dx.doi.org/10.7717/peerj.5674#supplemental-information.

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
