# Peer review of "Size dimorphism and sexual segregation in pheasants: tests of three competing hypotheses"

_PeerJ, doi:10.7717/peerj.5674_

## Round 0.1 · original submission · Major Revisions

Dear Authors,

I'm a new editor dedicated to handling your manuscript (as the previous Editor is unavailable), what takes me a few days more to read received referee comments and the MS. I generally like your work, but the comments by the referees are very useful and I guess not difficult to solve.

I suggest especially to re-think data handling and further analysis.
Moreover, because in the past I worked a little bit with SSD, I may suggest to include into introduction two more additional points; that if a difference between males and females is higher it probably helps better to explore foraging niches, although selection pressure can be in opposite selection due to other factors - see:

Tryjanowski, P., & Šimek, J. (2005). Sexual size dimorphism and positive assortative mating in red-backed shrike Lanius collurio: an adaptive value?. Journal of ethology, 23(2), 161-165.

and, I know is very speculative, may affect parasites community - lice are very important for phasanths, by the way:

Tryjanowski, P., Adamski, Z., Dylewska, M., Bulkai, L., & Rózsa, L. (2009). Demographic correlates of sexual size dimorphism and male genital size in the lice Philopterus coarctatus. Journal of Parasitology, 95(5), 1120-1124.

Reviewer 1 ·

Basic reporting

no comment

Experimental design

no comment

Validity of the findings

no comment

Additional comments

This manuscript reports the testing among several hypotheses that have been proposed (primarily based on studies in ungulates) to explain sexual segregation in the non-breeding season. Using pheasants hatched in captivity and then released to the wild, patterns of attendance at feeders relative to predator observations, gut contents of hunted birds, and activity patterns were assessed for individuals in their first year (prior to first reproduction) to provide data that should test among the three proposed hypotheses. The sample sizes are reasonable, though they found no support for any of the three tested hypotheses. However, each hypothesis does have a unique prediction that is testable, the authors are appropriately cautious in interpreting their results, and the manuscript is largely easy to follow.

1. I found it of interest that all of the hypotheses are explained, at least in part, by acquisition of food (males have exposure to greater predation risk to obtain higher quality food resources, males may need to exploit different food resources, males have greater activity due to need for greater energy acquisition). This might suggest that there could be many situations where sexual segregation may be due to a combination of all of the hypotheses, rather than just one (as the authors allude to) rather than to any single hypothesis. This raises the question about whether an analysis that might look at the hypotheses collectively could explain this behavior. I wonder if there might be more sophisticated analytical approaches that might look to see if a model that incorporates all of the data could potentially explain sexual segregation (I admit I am not certain whether there are appropriate analyses that would be suitable, but possibly some sort of structural equation modeling?).

2. No mention in either introduction or discussion is made about whether any aggression is known to occur between the sexes outside of the breeding season, either from published studies or from data collected as part of this study. This might be of interest to mention, as it could get at whether segregation is due to each sex preferentially seeking the same sex, or whether it was in part due to one sex excluding the other.

3. In the discussion, the authors raise some reasons why these hypotheses might be applicable to mammals, but I think this could also be raised in the introduction as a reason to evaluate this in a different type of taxa (e.g., taxa where much of the dimorphism is dichromatic, and less so for body size).

4. Lines 83-84: Do female birds have a smaller gut? Later on this is tested, but at this point you might indicate this is not known for birds (or if it is, what taxa and/or that it is variable across taxa).

5. I found some of the methods hard to follow, particularly the differences among years (they were raised slightly differently, data from some years used for some tests, etc.). Perhaps a table or flow chart that might include what was done with what birds raised in what year would help readers better follow what was done.

6. Lines 411-414: I think this is an important point that I am glad you brought up. I agree that they segregate at this stage, so this is only relevant to the hypotheses if selection for segregation at older ages drives segregations at this earlier age.

7. Line 435: In line with the previous comment, if aggregation behavior gets stronger as they age, then potentially future studies should evaluate this behavior in after hatch year birds rather than hatch year.

8. The discussion brings up the social preference hypothesis. However, from the introduction, it sounded as if there might be additional hypotheses that have been suggested to explain segregation. Although they were not tested, if these exist it might be nice to bring up in the discussion the ones that might be applicable to pheasants and discuss whether they are likely given the data and what is known about pheasants (or birds in general).

9. Figure 2 - why not show males versus females (since that is relevant to the hypothesis)?

10. Table 1: You already indicate that all of these hypotheses are for sexually dimorphic species, so I think that the column saying "yes" on this is not necessary. Indicating that you would not expect something to occur when taxa are monomorphic also seems unnecessary (particularly as this study ONLY included a species known to be sexually dimorphic, and you justify using that species partially due to that).

11. Table 1: The number of predictions based on information in the introduction is much longer than what is shown here. If these are just the predictions used in this study, then the monomorphic comments (given that the species is dimorphic) is not relevant. If it is more general, then I would include more in this table (possibly indicating those traits you studied).

12. Table 7: What does "Sig" mean? In my version, some of these values go on two lines and I am not clear what this means as the values are not what I might expect (e.g., in table 5, it is clear what Sig means as you indicate a p-value for each trait).

Reviewer 2 ·

Basic reporting

While the MS is generally well written throughout, there are a number of issues that could be improved, outlined below:
1. There are a number of different datasets used, in terms of birds sourced and raised under a range of different conditions. While I believe that this is handled OK during the analyses, it can at times be difficult to follow. I suggest putting the rearing conditions listed in the methods in under the methods for each hypothesis to be tested. That way the reader knows the history of the birds for a given hypothesis, without needing to recall what happened in year X.
2. The results could be improved through clarification of the language in regards to statistical outcomes. For example, lines 343-5, and again on line 364 and Line 372 - did these groups differ significantly? Line 353 what was the actual proportion here? Clarification for all results presented in this manner would improve readability immensely.
3. Line 488. Negative results can be difficult to interpret, but the newly proposed hypothesis should be dealt with prior to introduction during the conclusion section. Deal with likely interpretations, potential new hypotheses and so forth in sections prior to the conclusion, which should be a concise summary of the paper's findings.

Experimental design

This is comprehensive and a great use of multiple sources of data to test several hypotheses. No specific comments, looks sound.

Validity of the findings

No comment.

Additional comments

1. It would be helpful to know the scope of the project in terms of the monitoring methods. For example, we know how many birds were released and how many were detected, but this is not easily obtained for all birds. For example, Line 38 states 2400 birds released, then Line 40 that the authors 'determined their fate', however not all birds were recovered. It would be very useful to know how many individuals were detected on camera, observed in the field, measured gut etc for the relevant hypotheses at the start of the methods.
2. Line 50. This conclusion re: flight doesn't follow the data presented, and is particularly strange for a game bird that spends so long on the ground. Some of the most dimorphic species (e.g. large raptors) have superb flight capability, so this conclusion doesn't appear warranted based on data presented for mine.
3. Too often the paper refers to unpublished material, or another paper for key results. For example line 129 – how did social preferences drive this segregation? How natural were the rearing conditions? Line 175: the details are pertinent in this paper and should be summarised for the reader here, rather than requiring them to go to another paper for the information. I suggest that the authors search for all of their self-citations and include some base detail where the reference is not currently available, and/or it is key information for this paper.
4. Line 157. State the prediction here specifically.
5. Line 170-3. Be specific here: release notes lower numbers that was an unnaturally high density – which is it and what was the actual density to begin with? This whole paragraph could be reduced in length for the introduction – much of this is methods.
6. Line 315: Again, be specific, what distance, and how was it ascertained that this was not disturbing the birds?
7. Single sentence results with a new subheading for each is very difficult to read – suggest restructure.
8. Table 1 is not required and could be text. Table 2 could be supplemental material. Give degrees of Freedom for F statistic in Table 5.

Reviewer 3 ·

Basic reporting

No comment

Experimental design

No comment but see my General Commnents to the Author

Validity of the findings

No comment but see my General Commnents to the Author

Additional comments

This is a generally well written paper where authors try to disentangle the importance of various hypotheses to explain sexual segregation in the pheasant. My main concern is that all conclusions are based on artificially reared chicks, and thus the sexual segregation as adults could be influenced by a number of factors derived from the artificial setup. This artificial setup does not only affect the imprinting and socializing processes prior to release of the birds into the wild, but also the period during adulthood due to various causes, notably: (a) a possibly non-natural sex ratio at release could affect the behaviour of immature and adult birds when integrating into flocks; (b) the abnormally high density at release could also affect the behaviour of the birds in the wild; (c) the artificial feeders, and the perception of risky environments based on just the previous presence of foxes at them, ignoring raptors and hunting, does not emulate a natural situation.
At least I think the authors should discuss all these issues, and include a critical statement where they clearly admit that their study is based on artificially reared birds, and that the behaviour of released pheasants could have been critically influenced by the artificial setup and might not be extrapolable to a wild population.

A second main concern is that if the authors did not specifically test for the Social Preference Hypothesis in this study (see their own assertion in lines 497-498), then they cannot take that SPH is the most relevant as a conclusion of the current study.

Other important comments:

Introduction:

Second paragraph: There are more than three hypotheses to explain sexual segregation (Ruckstuhl & Neuhaus 2005 pp 15-19). Why do you reduce this number to three? Why not introducing here the Social Attraction and Social Avoidance Hypotheses as well? And what about other 'minor' or less studied hypotheses like the Weather Sensitivity or Physical Condition Hypotheses? If you exclude them here you should justify why.

lines 128-131: I do not agree that during adulthood social preferences may not be as important as other factors to explain sexual segregation.

lines 117, 128, 140, 176, 388, 435.....: I am not sure whether it is justified to cite here a submitted ms? There is no guarantee that this paper will be published, and I would suggest omitting reference to all these results from the present paper. The only section I would perhaps admit citation of this paper is the Discussion.

How could have the 8-10 week rearing period in captivity affected the biologically natural tendency to sexual segregation?

Methods:

line 199: Were the 30-individual groups mixed-sex groups? In which proportions were male and female chicks mixed in these groups? What was the sex-ratio at release? Please explain and discuss possible consequences of these factors for the social segregation hypothesis.

lines 206-207: What do you think could be the effect of feeding chicks on commercial chick crumbs on their possible sexually differential Forage Selection as adults?

Results:

To test the PRH you just used feeders, but you could have also used the time spent at other natural sites estimating the predation risk through other variables, e.g. presence of raptors, horizontal visibility of the foraging spots, etc.

lines 379-383: The Activity Budget Hypothesis test is not convincing. I can hardly believe that males and females spent the same time foraging/resting, and suspect that the absence of a difference is due to sample size and inappropriate parameters measured. For example, you could have compared the pecking rate of males and females, which I would expect to show some differences.

Discussion:

lines 391-393: Here you cite several examples of mammals, but since your study species is a bird, you should include some examples of birds showing SSD, at least great bustards which show the strongest SSD among birds (Alonso et al. 2009: The Auk 126:657-665), but maybe also others like González-Solis 2004: Oikos 105: 247-254, or the review by Payne 1984: Ornithological Monographs 33.

lines 398-406: These conclusions to me are not very convincing. First, the time taken to visit an artificial feeder is clearly not the best way to estimate risk-sensitivity. Second, it is very improbable that mortality rates do not differ between sexes in a species showing clear SSD.

lines 470-477: To exclude the Activity Budget Hypothesis as a major cause of sexual segregation, you only measured foraging, resting and walking, but not other social behaviours like interactions related to dominance and mating, which might also be very relevant in determining sexual segregation. I would expect males spending more time performing these activities than females. Thus, I am not sure that excluding the ABH as a relevant hypothesis is justified.

The conclusion about "avian systems" in lines 506-508 is not justified with just your example with pheasants. Other avian species show clear SSD and sexual segregation (great bustards are the most remarkable example, see e.g. Bravo et al. 2012: Bird Study 59: 243-251, Bravo et al 2016: The Auk 133: 178-197).

---

## Round 0.2 · Minor Revisions

I have read your responds to referee comments and I have decided to ACCEPT your manuscript, However I still believe that the Introduction can include some more general theoretical material. And because SSD is also important from parasites load point of view (or generally parasitology) which has practical implications in pheasant rearing, as well as in the poultry business. Please add one sentence on this into introduction and the MS will be ACCEPT soon.

---

## Round 0.3 · accepted · Accept

Well done! I have read personally the manuscript once again. Previously I decided only on minor revision, but now is the time to accept.

#